# Assessing the Suitability of Multivariate Analysis for Stress Tolerance Indices, Biomass, and Grain Yield for Detecting Salt Tolerance in Advanced Spring Wheat Lines Irrigated with Saline Water under Field Conditions

**Muhammad Mubushar** [1], **Salah El-Hendawy** [1,2,*], **Muhammad Usman Tahir** [1], **Majed Alotaibi** [1], **Nabil Mohammed** [1], **Yahya Refay** [1] and **ElKamil Tola** [3]

[1] Department of Plant Production, College of Food and Agriculture Sciences, King Saud University, Riyadh 11451, Saudi Arabia
[2] Department of Agronomy, Faculty of Agriculture, Suez Canal University, Ismailia 41522, Egypt
[3] Department of Agricultural Engineering, College of Food and Agriculture Sciences, King Saud University, Riyadh 11451, Saudi Arabia
[*] Correspondence: mosalah@ksu.edu.sa; Tel.: +966-5-3531-8364

**Abstract:** Successfully evaluating and improving the salt tolerance of genotypes requires an appropriate analysis tool to allow simultaneous analysis of multiple traits and to facilitate the ranking of genotypes across different growth stages and salinity levels. In this study, we evaluate the salt tolerance of 56 recombinant inbred lines (RILs) in the presence of salt-tolerant and salt-sensitive control genotypes using multivariate analysis of plant dry weight, measured at 75 (PDW-75) and 90 (PDW-90) days from sowing, biological yield (BY), grain yield (GY), and their salt tolerance indices (STIs). All RILs and genotypes were evaluated under the control and 15 dS m$^{-1}$ for two consecutive years (2019/2020 and 2020/2021). Results showed significant main effects of salinity and genotype as well as their interactions on four plant traits. Significant genotypic differences were also found for all calculated STIs. STIs exhibited moderate to strong relationships with the four plant traits when measured under either the control or salinity conditions and between each other. The principal component analysis (PCA) showed that the most variation among all analyzed variables was explained by the first two PCs, with the PC1 and PC2 explained at 61.8–71.8% and at 28.0–38.2% of the total variation, respectively. The PC1 had positive and strong correlations with the four plant traits measured under salinity conditions and STI, YI, REI, SWPI, MRPI, MPI, GMPI, and HMPI. The PC2 had strong correlations with BY and GY measured under the control conditions and SSI, TOL, RSE, and YSI. The PC1 was able to identify the salt-tolerant genotypes, while the PC2 was able to isolate the salt-sensitive ones. Cluster analysis based on multiple traits organized 64 genotypes into four groups varied from salt-tolerant to salt-sensitive genotypes, with the salt-tolerant group attaining higher value for plant traits under salinity conditions and the STIs related to the PC1. In conclusion, the use of multivariate analysis together with the STIs that evaluated the performance of genotypes under contrasting environmental conditions will help breeders to distinguish salt-tolerant genotypes from salt-sensitive ones, even at the early growth stages of plant development.

**Keywords:** biomass; bread wheat; grain yield; growth stages; principal component analysis; ranking; Ward's cluster analysis

## 1. Introduction

Today, the world is facing the worst food crisis, while drastic and unexpected changes in the climate are predicted to have a wide range of detrimental effects on global food security. This is because these changes are often accompanied by an increase in the intensity and frequency of several abiotic stresses. Such stresses would further threaten global food security because they are causing extensive yield losses in many food crops around

the world [1]. Therefore, addressing the negative impacts of abiotic stresses on agricultural production will offer a compelling contribution towards meeting the more than 60% increase in the demand for food required to feed 10 billion people by 2050 [2,3].

Among various abiotic stresses, salinity is the most common abiotic stress that significantly affects the productivity of several agricultural crops, particularly in arid and semiarid agricultural areas. In general, this stress affects about a billion hectares of land worldwide, of which approximately 33% (315.0 Mha) and 20% (191.0 Mha) is irrigated land and agricultural land, respectively, containing levels of salt high enough to substantially reduce the productivity of agricultural crops in these lands [4–6]. Furthermore, due to several reasons, which include excessive application of chemical fertilizers, inadequate amounts of precipitation, high temperature and surface evaporation, continuous irrigation with brackish water, intensive farming systems, and poor cultural practices, salinized lands continue to increase day-by-day and will account more than half of the world's agricultural land by 2050 [7,8]. Moreover, up to USD 30 billion is lost annually because the salinity stress can lead to approximately 30–50% yield losses [4,5,9]. For example, Qadir et al. [4] reported that, in rice cultivations, salinity stress causes a loss of up to USD 398 ha$^{-1}$ per year because it leads to a yield loss of 45%. Therefore, feasible and effective approaches are urgently needed to cope with the salinity problems in the agricultural sector.

Enhancing the salinity-tolerant ability of genotypes through a breeding and selection tool is seen as an effective and feasible approach to addressing the challenges of salinity problems in the agriculture sector and obtaining acceptable yields under salinity stress conditions [5,10,11]. Despite there are intensive efforts made by the research community to enhance the salt tolerance of genotypes, relatively few salt-tolerant genotypes have been released through breeding programs and even fewer have led to real-world applications. The variation of salt tolerance of genotypes with their growth stages, the lack of salinity studies related to the evaluation of salt tolerance of genotypes under actual saline field conditions and, until the yielding growth stage, and the lack of effective evaluation methods and screening criteria that have the potential to discriminate the salt tolerance among genotypes, are among the main factors limiting success in enhancing the salt tolerance of genotypes [5,10,12–14]. Additionally, several previous studies have reported that the combination between multiple selection criteria and appropriate statistical methods, that make the evaluation of stress tolerance convenient and efficient, are also required to succeed in breeding the stress-tolerant genotypes [14–18].

Previous studies have reported that determining the biomass yield over a long growth period could serve as an efficient criterion to assess the salt tolerance of genotypes at the different growth stages [19,20]. Additionally, because the final grain yield (GY) is always the main target of a plant breeder under both normal and stress conditions, the final GY could also be served as an essential criterion for evaluating the salt tolerance of genotypes. However, the performance of genotypes for the biomass and GY is often not consistent for all genotypes across stress and non-stress conditions. When different genotypes were evaluated under both conditions based on biomass and/or GY, the performance of genotypes can be classified into four groups: (1) genotypes with good performance in both conditions, (2) genotypes with weak performance in both conditions, (3) genotypes with good performance only in non-stress conditions, and (4) genotypes with good performance only in stress conditions [21–24]. Therefore, the most common approach to effectively identify tolerant genotypes is to select them on the basis of their biomass or GY performance in stress conditions relative to non-stress conditions [25–27]. To achieve this approach, different stress tolerance indices (STIs), which are calculated on a simple mathematical equation and based on the biomass and/or GY of each genotype under stress and non-stress conditions, have been proposed. Based on these indices, it is possible to determine the most tolerant and sensitive genotypes or identify desirable genotypes that perform well under both stress and non-stress conditions [14,15,28].

The yield stability index (YSI), stress tolerance index (STI), stress susceptibility index (SSI), tolerance index (TOI), yield index (YI), mean productivity index (MPI), geometric

mean productivity index (GMPI), mean relative performance index (MRPI), and relative efficiency index (REI) are examples of STIs that taking into account the observed values of the target trait of a genotype under stress and non-stress conditions, the mean values of the target trait of all genotypes under both conditions, as well as the variability in stress intensity over the environment and years [29–33]. The YSI is an effective index to assess the performance of a genotype under stress conditions relative to its performance under non-stress conditions and, therefore, it can be a suitable index for selecting the high stress-tolerant genotypes [34]. The STI is a suitable indicator for selecting the genotypes that have good performance under both stress and non-stress conditions and, therefore, the high values of this index indicate a high yield potential with a high tolerance of stress for a given genotype [30]. The SSI is a suitable index for isolating susceptible genotypes where the genotypes with higher SSI than the unit are more sensitive to stress, and vice versa [35]. The TOL is a suitable index to select the genotypes that have the least reduction in biomass or GY under stress conditions when compared with their values under non-stress conditions [29]. The YI is an effective index to identify a genotype with good performance under only stress conditions and failed to detect the genotypes with high performance under non-stress conditions [34]. The GMP, which is less sensitive to extreme values of biomass or grain yield, is a better index than MP, which has an upward bias when there are large differences between biomass or GY of stress and non-stress conditions, for identifying genotypes with good performance under non-stress conditions and reasonable performance under stress conditions [36].

Because many complicated polygenic crop traits control salinity tolerance, it is challenging to formulate conclusions concerning salt-tolerant and salt-sensitive genotypes on the basis of a specific trait or individual STI [26]. On the other hand, it is difficult to handle a large set of data from screening tests, as well as consider all STIs when evaluating the salt tolerance of a large number of genotypes without using appropriate statistical analysis. Therefore, for evaluating the salt tolerance of genotypes of a large number of genotypes based on a combination of STIs and different plant traits, there is a need for appropriate statistical tools to analyze multiple plant traits simultaneously and to be able to facilitate categorizing tested genotypes for salt tolerance [5,19]. Multivariate analysis (MA), including the principal component analysis (PCA) and cluster analysis (CA), is the most appropriate and successful analysis that makes the assessment of crop stress performance more practical and reliable, identifies the most important plant traits governing stress tolerance, and discriminates the most stress-tolerant genotypes when the screening of genotypes for stress tolerance was made on multiple traits, different salinity levels, and different growth stages [11,37,38]. Additionally, regression analysis is also used as a useful tool for identifying the different adaptation mechanisms to stress conditions. This tool was successful to classify different barley genotypes based on GY under drought stress conditions into three different groups. The first group included the genotypes that produced a high GY under non-stress conditions but showed a drastic reduction in GY under drought stress conditions. The second group included genotypes that had a lower GY in the non-stress conditions, but the reduction in their GY in drought stress conditions was not as significant as the genotypes of the first group, and the third group included genotypes that had stable yield under both conditions [23].

The main objective of this study was to evaluate a large number of wheat genotypes, including advanced breeding lines and commercial cultivars, for salt tolerance potential under saline field conditions based on plant biomass detected at different growth stages and GY. The specific objectives were to (1) identify the relative importance of different STIs in evaluating the salt tolerance of wheat genotypes at different growth stages, (2) use MA to accurately facilitate the evaluation process of salt tolerance among different genotypes using multiple and various traits, and (3) use cluster analysis numbers as a simple way to rank the genotypes for their salt tolerance at different growth stages and, finally, classify them into different categories based on their salt tolerance regardless of the stage of growth.

The results of this study could provide the identification of suitable wheat genotypes which can be successfully used in wheat production on less productive and saline farmers' fields.

## 2. Materials and Methods

### 2.1. Plant Materials and Experimental Site Conditions

This study was carried out with a wide range of genetic variability (sixty-four bread wheat genotypes) and two levels of salinity stress. The tested genotypes consisted of two groups of developed genetic materials (F8 recombinant inbred lines, RILs), three parents (Sakha 61, Sids 1, and Sakha 93), and five commercial cultivars (Shandawel 1, Gemiza 9, Misr 1, Kharchia 65, and Kawz). The parents and commercial cultivars were previously evaluated under salinity conditions and Sakha 61 and Shandawel-1 have been identified as salt-sensitive, Misr-1 has been identified as moderately salt-sensitive, Sids 1 and Gemiza 9 have been identified as moderately salt-tolerant, and Sakha 93 and Kharchia 65 have been identified as salt-tolerant cultivars. [13,19,39,40]. The first group of RILs (28 RILs) was developed from a cross between Sakha 93 and Sakha 61, while the second one (28 RILs) was developed from a cross between Sakha 93 and Sids 1.

These plant materials were evaluated during two growing seasons (2019/2020 and 2020/2021) under open field conditions at the experimental farm of the College of Food and Agriculture Sciences, King Saud University, Riyadh, Saudi Arabia (24°25′ N, 46°34′ E, 400 m a.s.l.) using normal water ($\approx$0.35 dS m$^{-1}$) and high artificial saline water ($\approx$15.0 dS m$^{-1}$) (Figure 1). This experimental area is characterized by arid conditions, where the summer season (May to October) is with very hot and dry days and temperatures can be reached 50 °C, while the winter season (November to April) is mostly sunny with mean temperature and rainfall varying from 12.9 to 32.2 °C and 4.0 to 20.0 mm, respectively. The soil texture of the experimental farm is sandy loam (56.7% sand, 28.4% silt, and 14.9% clay) with a pH, organic matter, bulk density, and electrical conductivity (EC) of 7.85, 0.46%, 1.48 g cm$^{-3}$, and 1.12 dS m$^{-1}$, respectively. Additionally, the availability of N, $K_2O$, and $P_2O_5$ in the soil was 3.98, 1.67, and 0.07 g kg$^{-1}$ dry soil, respectively [41].

### 2.2. Experimental Design, Agronomic Practices, and Salinity Treatments

Field experiments were conducted using a split-plot design with three replications. The salinity treatments (the control and a high salinity level of 15.0 dS m$^{-1}$) were arranged in the main plot, while the genotypes were randomly arranged in the subplots. After the experimental site was plowed and leveled, it was divided into two main plots: one for the control and another for salinity stress. Each main plot was divided into 16 main subplots at a size of 4 m × 7.5 m for each and a 1 m buffer zone between them. Thereafter, each main subplot consisted of 12 small subplots at a size of 1 m × 1.5 m, spaced 50 cm apart (Figure 1). The genotypes were randomly distributed on these subplots and the seeds of each genotype were planted in five 1.5 m-long rows spaced 20 cm apart (1.5 m$^2$ in total area) on 25 November 2019 and 17 November 2020, at a seeding rate of 150 kg ha$^{-1}$ in both the control and salinity treatments. Each genotype was also fertilized at a rate of 150 kg N ha$^{-1}$, 100 kg $P_2O_5$ ha$^{-1}$, and 90 kg $K_2O$ ha$^{-1}$. Other agronomic practices, such as pest and disease control, were performed as recommended by the local region of the experimental field.

Initially, the genotypes in both treatments were irrigated with normal water up to 21 days after seeding to avoid any osmotic shock and facilitate germination and early seedling establishment. Thereafter, the genotypes in the control treatment continued to be irrigated with normal water, while in salinity treatment, they were irrigated with artificial saline water containing a 150 mM NaCl L$^{-1}$ solution. The artificial saline water was prepared by adding 8.8 g NaCl L$^{-1}$ and mixed well in a separate plastic tank (5.0 m$^3$) to achieve the target salinity level. The artificial saline water was delivered for each subplot via a low-pressure surface irrigation system. This system consisted of a main line (76 mm in diameter), which connected with the plastic tank, branched off at each main subplot to sub-main hoses, and equipped with a manual control valve (Figure 1). The same

irrigation system was used to deliver non-saline water for the control treatment. The irrigation frequency and quantity of water applied for each irrigation event were adjusted according to environmental conditions and plant phenology. Additionally, the salinity level was constantly maintained during the growing season and both years through collected soil samples at a depth of 0–60 cm from different places of the main plot, and their EC was measured.

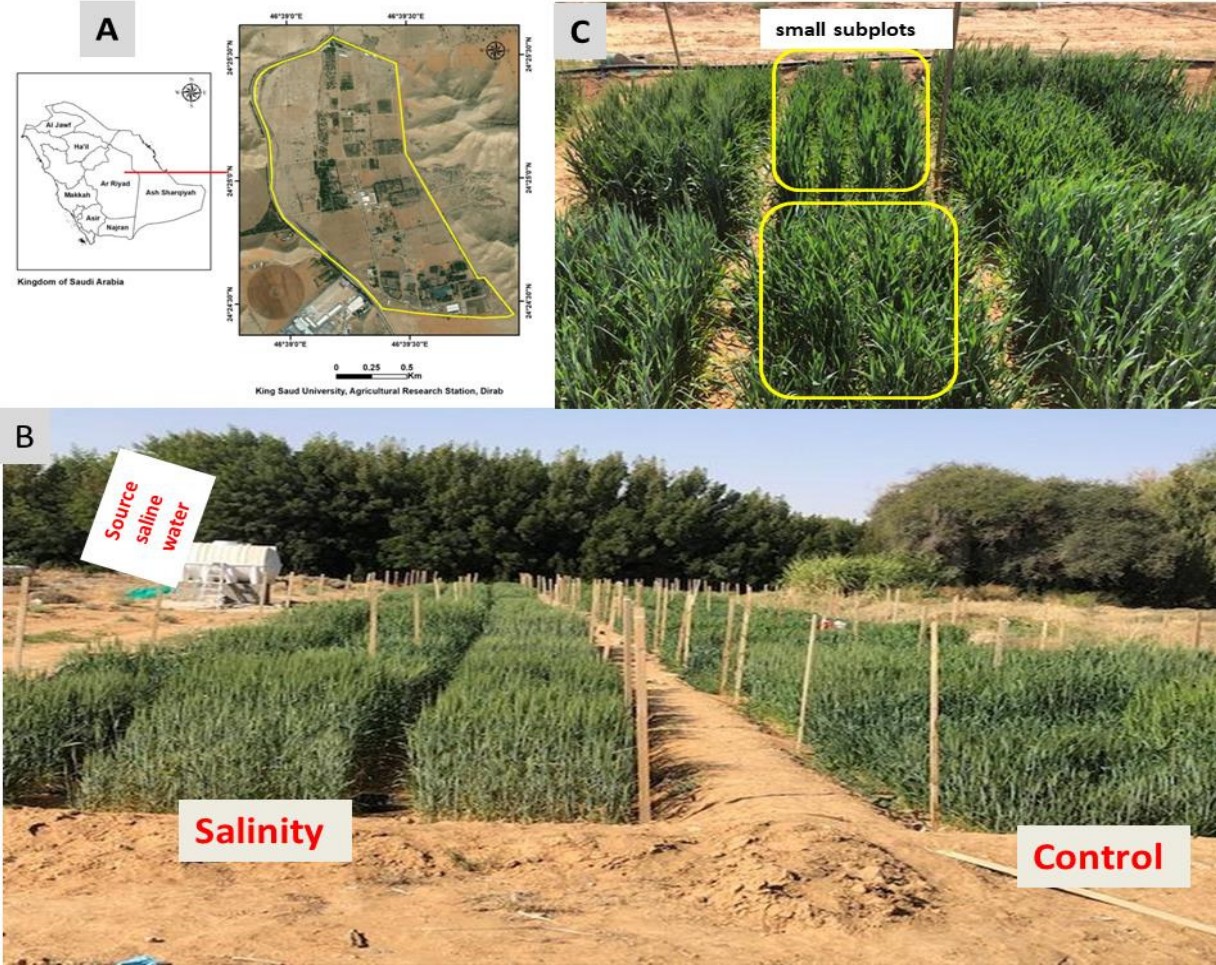

**Figure 1.** Location map of the experimental field (**A**), plot layouts of the control and salinity treatments (**B**), and the subplot size of the experimental field (**C**).

*2.3. Measurements*

2.3.1. Determination of Plant Traits

At Zadoks scale 49 and 65, which refer to booting (about at 75 days from sowing) and anthesis (about at 90 days from sowing) wheat growth stages [42], respectively, ten plants from each genotype in each treatment and replication were selected at random, divided into different parts (leaves, stems, and spikes), and then the separated parts were oven dried at 70 °C to a constant weight and weighed using a digital balance with an accuracy of ±0.001 g. Finally, the total plant dry weight (PDW) at both stages, which is referred to as a PDW-75 and PDW-90, was determined as the sum of the dry weight of the three parts.

At Zadoks scale 92, which refers to the ripening wheat stage (at about 150 days from sowing), biological yield (BY) and GY were determined by hand harvesting an area of three internal rows, each 1.25 m in length (0.75 m$^2$ in total area) from each genotype. The plants were air-dried for 7 days and weighed to determine the BY. Next, the spikes were separated and the grains were collected, cleaned, adjusted to a moisture content of 14.0%, and weighed to determine the GY. The BY and GY were finally expressed as a ton ha$^{-1}$.

2.3.2. Calculation of Different STIs

The different STIs were calculated for each genotype based on the values of PDW-75, PDW-90, BY, and GY under the control and salinity conditions. The full name, abbreviation, formula, and reference of each STI is summarized in Table 1.

**Table 1.** The abbreviations (Abb.), full index name, formula, and references of the different stress tolerance indices (STIs) used in this study.

| Abb. | Index Name | Formula | Reference |
|------|-----------|---------|-----------|
| YSI | Yield stability index | $S/NS$ | [34] |
| TOL | Tolerance index | $NS - S$ | [29] |
| YI | Yield index | $S/$ | [43] |
| STI | Stress tolerance index | $(NS \times S)/(\text{N})^2$ | [30] |
| SSI | Stress susceptibility index | $1 - (S/NS)/1 - (/\text{N})$ | [35] |
| MPI | Mean productivity index | $(NS + S)/2$ | [29] |
| GMPI | Geometric mean productivity index | $\sqrt{NS \times S}$ | [30] |
| MRPI | Mean relative performance index | $(S/) + (NS/\text{N})$ | [36] |
| REI | Relative efficiency index | $(S/) \times (NS/\text{N})$ | [35] |
| HMPI | Harmonic mean productivity index | $(2 \times (NS \times S))/(NS + S)$ | [44] |
| SWPI | Stress-weighted performance index | $S/\sqrt{NS}$ | [45] |
| RSE | Relative salinity effect | $(NS - S)/NS \times 100$ | [46] |

$S$ and $NS$ are the values of the trait of each genotype evaluated under stress and non-stress conditions, respectively. $\acute{S}$ and $N$ are the mean values of all genotypes evaluated under stress and non-stress conditions, respectively.

*2.4. Analysis of Data*

The following steps of data analysis were performed on PDW-75, PDW-90, BY, GY, and different STIs, in order to detect the salt tolerance of the evaluated wheat genotypes. To assess the impact of salinity, genotype, and their interaction on the different plant traits, the values of each year were subjected to analysis of variance (ANOVA) appropriate for a split-plot design according to a completely randomized design. Prior to analysis, the normality of each plant trait was tested using the Shapiro–Wilk test. A box plot was used to present the descriptive statistics of the four traits. Pearson's correlation matrix was used to estimate the level of correlation between plant traits and different STIs and between each other. To identify the traits that contributed to most of the variation in tested wheat genotypes, to detect the interrelationships among multiple traits of each other, and to identify which genotypes were more tolerant or sensitive to salinity stress, the PCA was applied to the genotype by a trait matrix of means and a biplot was drawn using the XLSTAT package. To group genotypes according to their level of salt tolerance, a hierarchical cluster analysis was performed at each growth stage based on different STIs, and plant traits were measured under the control and salinity conditions. The analysis was performed according to Ward's method, where the distances between the two clusters were expressed as squared Euclidean distances. A dendrogram of clusters was performed to identify the cluster groups, whereas the K-means analysis was used to identify the number of clusters. The ranking of genotypes for salt tolerance at each growth stage and across different growth stages was performed according to the methods of El-Hendawy et al. [19].

**3. Results**

*3.1. Analysis of Variance (ANOVA)*

The F-values in the ANOVA analysis showed that salinity (S), genotype (G), and their interaction had a highly significant effect ($p < 0.001$) on all traits measured at different growth stages (PDW-75, PDW-90, BY, and GY) in each year and the combined analysis of two years (Table 2). All traits were not affected by year (Y), while their interaction with S had a significant effect ($p < 0.05$) on PDW-75 and PDW-90, and their interaction with G had a highly significant effect ($p < 0.001$) on all traits. All traits were not affected by the three-way interaction (G × S × Y, Table 2).

**Table 2.** Analysis of variance (F-values) for plant dry weight measured at 75 (PDW-75) and 90 (PDW-90) days after sowing, biological yield (BY), and grain yield (GY) of 64 wheat genotypes for each year and the combined analysis of two years.

| Source of Variance | DF | PDW-75 | PDW-90 | BY | GY |
|---|---|---|---|---|---|
| | | **First Year** | | | |
| Salinity (S) | 1 | 8374.7 *** | 8710.8 *** | 3415.9 *** | 12,071.1 *** |
| Genotype (G) | 63 | 22.34 *** | 34.43 *** | 5.71 *** | 11.79 *** |
| G × S | 63 | 4.91 *** | 8.61 *** | 2.10 *** | 4.90 *** |
| | | **Second year** | | | |
| Salinity (S) | 1 | 1610.3 *** | 7420.6 *** | 2487.1 *** | 5055.9 *** |
| Genotype (G) | 63 | 12.87 *** | 22.48 *** | 9.75 *** | 7.64 *** |
| G × S | 63 | 2.10 *** | 6.28 *** | 3.45 *** | 3.12 *** |
| | | **Combined two years** | | | |
| Year (Y) | 1 | 11.03 ns | 0.63 ns | 30.73 ns | 17.70 ns |
| Salinity (S) | 1 | 5698.4 *** | 16,078.4 *** | 5753.1 *** | 14,064.2 *** |
| S × Y | 1 | 8.89 * | 8.34 * | 0.024 ns | 3.66 ns |
| Genotype (G) | 63 | 28.49 *** | 48.09 *** | 13.43 *** | 16.50 *** |
| G × Y | 63 | 4.62 *** | 6.85 *** | 1.79 *** | 1.97 *** |
| G × S | 63 | 5.77 *** | 13.97 *** | 5.08 *** | 7.33 *** |
| G × S × Y | 63 | 0.61 ns | 0.52 ns | 0.38 ns | 0.28 ns |

\* $p \leq 0.05$, \*\*\* $p \leq 0.001$, ns: not significant.

The F-values in each year and the combined analysis of two years also showed a high genetic variation among genotypes for all STIs that were calculated based on PDW-75, PDW-90, BY, or GY (Table 3). All STIs were not affected by Y, except for YI, SSI, and TOL calculated based on GY. The Y interactions with G significantly influenced all STIs calculated based on PDW-75 and all STIs calculated based on PDW-90, except YSI, SSI, TOL, and RSE, while this interaction effect was insignificant for all STIs calculated based on BY or GY (Table 3).

**Table 3.** Analysis of variance (F-values) for different stress tolerance indices (STIs) calculated based on plant dry weight measured at 75 (PDW-75) and 90 (PDW-90) days after sowing, biological yield (BY), and grain yield (GY) of 64 wheat genotypes for the first year (Y1), second year (Y2), and combined across two years (C).

| Source | DF | YSI | STI | YI | REI | SWPI | MRPI | MPI | GMPI | HMPI | SSI | TOL | RSE |
|---|---|---|---|---|---|---|---|---|---|---|---|---|---|
| | | **STIs calculated based on PDW-75** | | | | | | | | | | | |
| G (Y1) | 63 | 11.47 ** | 12.61 ** | 14.44 ** | 12.58 ** | 13.93 ** | 14.40 ** | 14.20 ** | 14.23 ** | 14.26 ** | 11.84 ** | 11.54 ** | 11.88 ** |
| G (Y2) | 63 | 11.01 ** | 6.62 *** | 9.56 *** | 6.64 *** | 12.22 ** | 7.76 *** | 7.38 *** | 7.64 *** | 7.92 *** | 10.99 ** | 8.15 *** | 11.05 ** |
| G (C) | 63 | 20.57 ** | 14.56 ** | 20.20 ** | 14.67 ** | 23.41 ** | 17.48 ** | 16.98 ** | 17.43 ** | 17.92 ** | 20.96 ** | 17.89 ** | 21.05 ** |
| G × Y | 63 | 1.94 *** | 2.72 *** | 2.97 *** | 2.72 *** | 2.74 *** | 2.82 *** | 2.75 *** | 2.76 *** | 2.76 *** | 1.90 *** | 1.89 *** | 1.94 *** |
| | | **STIs calculated based on PDW-90** | | | | | | | | | | | |
| G (Y1) | 63 | 14.50 ** | 22.67 ** | 23.21 ** | 23.28 ** | 18.76 ** | 23.97 ** | 23.63 ** | 23.87 ** | 24.08 ** | 14.22 ** | 15.94 ** | 14.28 ** |
| G (Y2) | 63 | 15.38 ** | 13.94 ** | 13.49 ** | 14.17 ** | 14.96 ** | 13.55 ** | 13.67 ** | 13.44 ** | 13.29 ** | 15.86 ** | 17.77 ** | 15.86 ** |
| Y | 1 | 2.89 ns | 16.63 ns | 4.00 ns | 0.41 ns | 10.10 ns | 1.49 ns | 0.63 ns | 0.11 ns | 0.02 ns | 2.93 ns | 3.31 ns | 3.96 ns |
| G (C) | 63 | 28.95 ** | 29.54 ** | 31.22 ** | 30.31 ** | 31.59 ** | 30.73 ** | 30.70 ** | 30.53 ** | 30.58 ** | 29.16 ** | 32.41 ** | 29.17 ** |
| G × Y | 63 | 0.89 ns | 4.82 *** | 3.76 *** | 4.90 *** | 2.02 *** | 4.42 *** | 4.37 *** | 4.40 *** | 4.34 *** | 0.88 ns | 1.21 ns | 0.88 ns |
| | | **STIs calculated based on BY** | | | | | | | | | | | |
| G (Y1) | 63 | 9.10 *** | 3.51 *** | 6.07 *** | 3.53 *** | 9.34 *** | 3.70 *** | 3.45 *** | 3.78 *** | 4.21 *** | 9.20 *** | 6.11 *** | 9.19 *** |
| G (Y2) | 63 | 12.41 ** | 6.16 *** | 9.53 *** | 6.19 *** | 13.14 ** | 6.50 *** | 6.11 *** | 6.70 *** | 7.33 *** | 12.41 ** | 8.53 *** | 12.48 ** |
| Y | 1 | 4.74 ns | 12.21 ns | 0.004 ns | 0.15 ns | 1.94 ns | 0.004 ns | 3.67 ns | 3.39 ns | 3.71 ns | 2.88 ns | 0.06 ns | 3.16 ns |
| G (C) | 63 | 20.49 ** | 8.31 *** | 14.16 ** | 8.34 *** | 21.31 ** | 8.79 *** | 8.25 *** | 9.13 *** | 10.16 ** | 20.66 ** | 13.65 ** | 20.65 ** |
| G × Y | 63 | 1.01 ns | 1.09 ns | 1.16 ns | 1.06 ns | 1.22 ns | 1.06 ns | 1.10 ns | 1.14 ns | 1.17 ns | 1.10 ns | 1.03 ns | 1.03 ns |
| | | **STIs calculated based on GY** | | | | | | | | | | | |
| G (Y1) | 63 | 11.52 ** | 6.99 *** | 6.61 *** | 6.92 *** | 8.17 *** | 6.92 *** | 7.32 *** | 7.04 *** | 6.89 *** | 11.42 ** | 12.57 ** | 11.30 ** |
| G (Y2) | 63 | 11.12 ** | 4.73 *** | 5.55 *** | 4.69 *** | 8.06 *** | 4.53 *** | 4.49 *** | 4.57 *** | 4.75 *** | 11.67 ** | 10.59 ** | 11.74 ** |
| Y | 1 | 5.85 ns | 9.09 ns | 0.92 * | 1.53 ns | 14.35 ns | 0.02 ns | 17.60 ns | 16.61 ns | 15.82 ns | 40.07 * | 34.8 * | 6.78 ns |
| G (C) | 63 | 21.75 ** | 10.10 ** | 10.93 ** | 10.01 ** | 15.27 ** | 9.80 *** | 9.90 *** | 9.77 *** | 9.89 *** | 22.20 ** | 22.13 ** | 22.14 ** |
| G × Y | 63 | 0.88 ns | 1.18 ns | 1.06 ns | 1.20 ns | 0.94 ns | 1.17 ns | 1.18 ns | 1.20 ns | 1.21 ns | 0.89 ns | 0.83 ns | 0.88 ns |

\* $p \leq 0.05$, \*\* $p \leq 0.01$, \*\*\* $p \leq 0.001$, ns: not significant. The full names of the different STIs are listed in Table 1.

### 3.2. Genotypic Variability of Traits and STIs under the Control and Salinity Stress Conditions

The destructive statistics of the traits and STIs for all genotypes under the control and salinity stress conditions are presented in Figure 2 (as a box plot) and Table 4, respectively. In general, averaged over the two years, salinity stress significantly reduced PDW-75, PDW-90, BY, and GY by 30.3, 33.1, 30.0, and 32.1%, respectively, as compared to the control conditions (Figure 2). About a two-fold variation was found in the four traits among the genotypes under the control and stress conditions, with PDW-75, PDW-90, BY, and GY ranging from 3.81 to 8.06 g plant$^{-1}$, 6.12 to 11.54 g plant$^{-1}$, 14.53 to 23.73 t ha$^{-1}$, and 3.97 to 7.09 t ha$^{-1}$ under the control conditions, and from 2.46 to 5.48 g plant$^{-1}$, 3.49 to 7.60 g plant$^{-1}$, 8.90 to 17.22 t ha$^{-1}$, and 2.77 to 4.93 t ha$^{-1}$ under the salinity stress conditions, respectively (Figure 2).

Similar to the four traits, the mean values of different STIs also varied significantly among the 64 genotypes (Table 4). The maximum values of the YSI, YI, MRPI, MPI, GMPI, stress-weighted performance index (SWPI), and harmonic mean productivity index (HMPI) were about two times higher than those of the minimum values, while the maximum values of the STI, REI, SSI, TOL, and relative salinity effect (RSE), were about 2 to 4 times higher than those of the minimum values (Table 4).

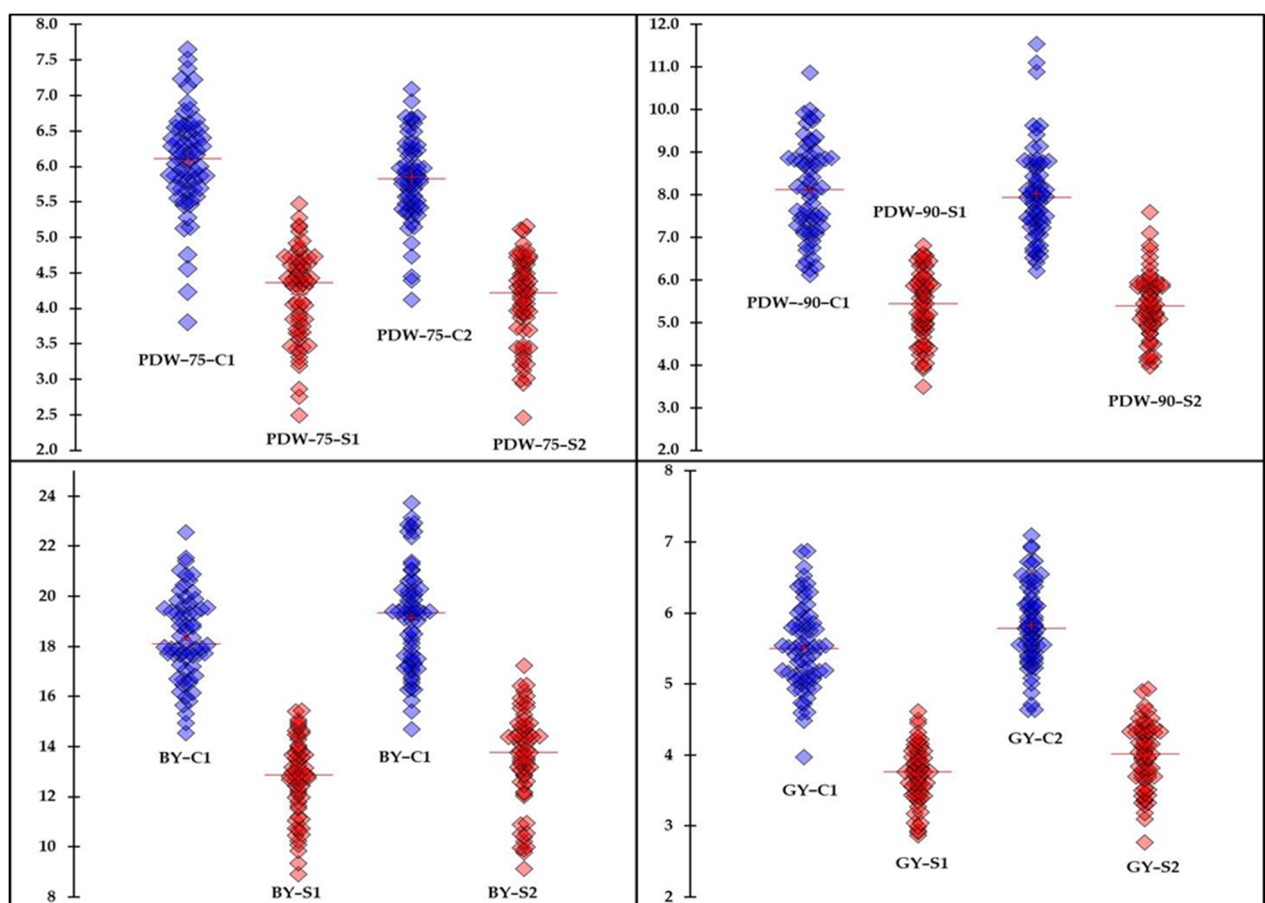

**Figure 2.** Box plots showing the variability for plant dry weight measured at 75 (PDW-75) and 90 (PDW-90) days after sowing, biological yield (BY), and grain yield (GY) of 64 wheat genotypes under the control (C) and salinity (S) conditions for the first year (Y1) and second year (Y2).

**Table 4.** Destructive statistics ((minimum (Min), maximum (Max), range, mean values, and variation between Max and Min values (Var.)) of the different stress tolerance indices (STIs) calculated based on plant dry weight measured at 75 (PDW-75) and 90 (PDW-90) days after sowing, biological yield (BY), and grain yield (GY) of 64 wheat genotypes. Data are the average of two years and three replications (*n* = 6).

| Source | YSI | STI | YI | REI | SWPI | MRPI | MPI | GMPI | HMPI | SSI | TOL | RSE |
|---|---|---|---|---|---|---|---|---|---|---|---|---|
| | STIs based on PDW-75 | | | | | | | | | | | |
| Min | 0.53 | 0.30 | 0.60 | 0.44 | 1.19 | 1.32 | 3.41 | 3.28 | 3.15 | 0.57 | 1.06 | 17.36 |
| Max | 0.83 | 1.02 | 1.27 | 1.47 | 2.09 | 2.43 | 6.22 | 6.02 | 5.82 | 1.56 | 3.15 | 47.20 |
| Range | 0.30 | 0.72 | 0.68 | 1.03 | 0.90 | 1.10 | 2.81 | 2.74 | 2.67 | 0.98 | 2.09 | 29.84 |
| Mean | 0.70 | 0.71 | 1.00 | 1.01 | 1.70 | 2.00 | 5.06 | 4.97 | 4.88 | 1.00 | 1.81 | 30.18 |
| Var. (times) | **1.6** | **3.4** | **2.1** | **3.4** | **1.8** | **1.8** | **1.8** | **1.8** | **1.8** | **2.7** | **3.0** | **2.7** |
| | STIs based on PDW-90 | | | | | | | | | | | |
| Min | 0.41 | 0.40 | 0.71 | 0.60 | 1.32 | 1.54 | 5.25 | 5.08 | 4.89 | 0.53 | 1.25 | 17.66 |
| Max | 0.82 | 1.12 | 1.25 | 1.67 | 2.29 | 2.58 | 8.73 | 8.50 | 8.27 | 1.79 | 6.11 | 59.07 |
| Range | 0.41 | 0.72 | 0.54 | 1.08 | 0.97 | 1.03 | 3.49 | 3.41 | 3.38 | 1.25 | 4.86 | 41.40 |
| Mean | 0.67 | 0.68 | 1.00 | 1.01 | 1.90 | 2.00 | 6.74 | 6.59 | 6.45 | 0.99 | 2.67 | 32.63 |
| Var. (times) | **2.0** | **2.8** | **1.8** | **2.8** | **1.7** | **1.7** | **1.7** | **1.7** | **1.7** | **3.3** | **4.9** | **3.3** |
| | STIs based on BY | | | | | | | | | | | |
| Min | 0.48 | 0.43 | 0.70 | 0.62 | 2.12 | 1.58 | 12.82 | 12.34 | 11.83 | 0.49 | 2.72 | 14.70 |
| Max | 0.85 | 0.97 | 1.21 | 1.38 | 3.66 | 2.35 | 18.63 | 18.40 | 18.17 | 1.75 | 10.43 | 52.47 |
| Range | 0.38 | 0.53 | 0.51 | 0.76 | 1.55 | 0.76 | 5.82 | 6.06 | 6.35 | 1.26 | 7.71 | 37.77 |
| Mean | 0.70 | 0.71 | 1.00 | 1.01 | 3.03 | 2.00 | 15.94 | 15.66 | 15.38 | 0.99 | 5.62 | 29.64 |
| Var. (times) | **1.8** | **2.2** | **1.7** | **2.2** | **1.7** | **1.5** | **1.5** | **1.5** | **1.5** | **3.6** | **3.8** | **3.6** |
| | STIs based on GY | | | | | | | | | | | |
| Min | 0.50 | 0.45 | 0.74 | 0.67 | 1.25 | 1.63 | 3.87 | 3.79 | 3.64 | 0.54 | 0.86 | 17.23 |
| Max | 0.83 | 0.98 | 1.22 | 1.45 | 1.89 | 2.40 | 5.70 | 5.59 | 5.49 | 1.55 | 3.08 | 49.76 |
| Range | 0.33 | 0.53 | 0.48 | 0.78 | 0.64 | 0.77 | 1.83 | 1.81 | 1.86 | 1.01 | 2.21 | 32.53 |
| Mean | 0.68 | 0.68 | 1.00 | 1.01 | 1.62 | 2.00 | 4.76 | 4.66 | 4.57 | 0.99 | 1.82 | 31.69 |
| Var. (times) | **1.6** | **2.2** | **1.7** | **2.2** | **1.5** | **1.5** | **1.5** | **1.5** | **1.5** | **2.9** | **3.6** | **2.9** |

The full names of the different STIs are listed in Table 1.

### 3.3. Traits and Genotypes Association

#### 3.3.1. Correlation Analysis

The correlation analysis showed that different STIs were either positively or negatively correlated with the four plant traits and between themselves (Figure 3). The STIs which had a strong and positive correlation with the plant trait measured under the control or salinity stress conditions were STI, REI, MRPI, MPI, GMPI, and HMPI (r = 0.81–0.95). These aforementioned STIs did not exhibit any significant correlations with the TOL calculated based on PDW-75 and PDW-90, SSI and RSE calculated based on BY, and SSI, TOL, and RSE calculated based on GY, whereas they showed a moderate-to-strong positive correlation with SWPI calculated based on the four traits (r = 0.40–0.82) and a moderate negative correlation with SSI and RSE calculated based on PDW-75 and PDW-90 (r ranged from −0.29 to −0.45). The following STIs: YSI, YI, SWPI, SSI, and RSE, correlated better with the four traits under salinity stress than those under the control conditions; the opposite was true for TOL. The TOL, SSI, and RSE calculated based on the four traits did not show any significant correlations with the STIs; they showed strong and negative correlations with only YSI and SWPI (Figure 3).

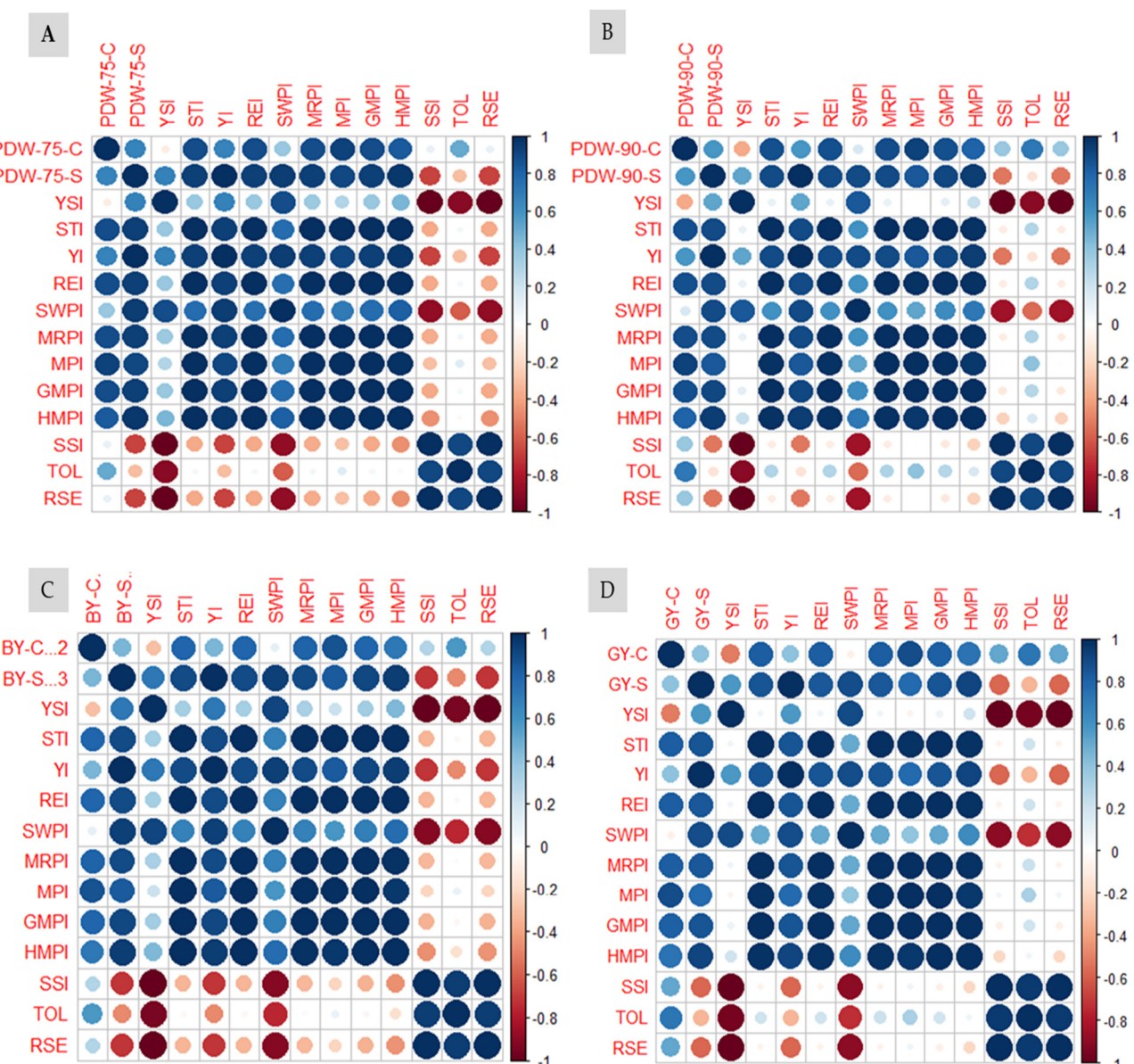

**Figure 3.** Pearson's correlation coefficients between different stress tolerance indices and (**A**) plant dry weight measured at 75 days after sowing (PDW-75), (**B**) plant dry weight measured at 90 days after sowing (PDW-90), (**C**) biological yield (BY), and (**D**) grain yield (GY) of 64 wheat genotypes grown under the control (C) and salinity stress (S) conditions. The strength of the correlation between the two traits is based on the size and color of the circle. The dark blue and dark red of the color scale reflects a completely positive and negative correlation between the two traits. The full names of the different STIs are listed in Table 1.

### 3.3.2. Principal Component Analysis

Results of the PCA indicated that the first two components had an Eigenvalue > 1 and explained about 99.9% of the total variation among all analyzed variables (Table 5 and Figure 4). The first component (PC1) explained 71.8, 64.6, 69.2, and 61.8% of the total variation among the 64 genotypes assessed by PDW-75, PDW-90, BY, and GY measured under the control and salinity conditions, as well as the different STIs, and the second component (PC2) explained 28.0, 35.1, 30.7, and 38.2% of the total variation assessed by the same variables, respectively (Table 5 and Figure 4). Additionally, the PC1 had a strong positive correlation with PDW-75 and PDW-90 measured under the control and salinity

conditions, BY and GY measured under salinity conditions, STI, YI, REI, SWPI, MRPI, MPI, GMPI, and HMPI calculated based on any plant trait. The PC1 had a moderate positive correlation with YSI and a moderate negative correlation with SSI and RSE calculated based on PDW-75 and BY (Table 5).

**Table 5.** Eigenvalue, cumulative variability, and factor loadings of the first two principal components (PCs) for plant dry weight measured at 75 days after sowing (PDW-75), plant dry weight measured at 90 days after sowing (PDW-90), biological yield (BY), and grain yield (GY) of 64 wheat genotypes grown under the control (C) and salinity stress (S) conditions.

| Traits | PDW-75 | | PDW-90 | | BY | | GY | |
|---|---|---|---|---|---|---|---|---|
| | **PC1** | **PC2** | **PC1** | **PC2** | **PC1** | **PC2** | **PC1** | **PC2** |
| PDW-C | **0.708** | **0.704** | **0.752** | **0.656** | | | | |
| PDW-S | **0.998** | −0.057 | **0.974** | −0.225 | | | | |
| BY-C | | | | | 0.532 | **0.846** | | |
| BY-S | | | | | **0.996** | −0.091 | | |
| GY-C | | | | | | | 0.636 | **0.770** |
| GY-S | | | | | | | **0.962** | −0.270 |
| YSI | 0.631 | **−0.775** | 0.318 | **−0.947** | 0.646 | **−0.762** | 0.335 | **−0.942** |
| STI | **0.953** | 0.295 | **0.973** | 0.222 | **0.933** | 0.356 | **0.961** | 0.274 |
| YI | **0.998** | −0.057 | **0.974** | −0.225 | **0.996** | −0.091 | **0.963** | −0.268 |
| REI | **0.953** | 0.295 | **0.973** | 0.221 | **0.934** | 0.356 | **0.960** | 0.274 |
| SWPI | **0.920** | −0.390 | **0.772** | −0.635 | **0.896** | −0.444 | **0.722** | −0.692 |
| MRPI | **0.954** | 0.299 | **0.973** | 0.230 | **0.931** | 0.365 | **0.961** | 0.277 |
| MPI | **0.926** | 0.376 | **0.944** | 0.327 | **0.886** | 0.463 | **0.919** | 0.393 |
| GMPI | **0.957** | 0.290 | **0.978** | 0.208 | **0.939** | 0.344 | **0.965** | 0.263 |
| HMPI | **0.977** | 0.209 | **0.995** | 0.092 | **0.972** | 0.233 | **0.990** | 0.135 |
| SSI | −0.630 | **0.775** | −0.319 | **0.947** | −0.646 | **0.762** | −0.335 | **0.942** |
| TOL | −0.247 | **0.966** | 0.081 | **0.993** | −0.388 | **0.920** | −0.069 | **0.996** |
| RSE | −0.631 | **0.775** | −0.318 | **0.947** | −0.646 | **0.762** | −0.335 | **0.942** |
| | PDW-75 | | PDW-90 | | BY | | GY | |
| Eigenvalue | 10.05 | 3.92 | 9.05 | 4.92 | 9.69 | 4.29 | 8.64 | 5.34 |
| Variability (%) | 71.79 | 28.02 | 64.63 | 35.13 | 69.22 | 30.66 | 61.75 | 38.12 |
| Cumulative (%) | 71.79 | 99.81 | 64.63 | 99.76 | 69.22 | 99.88 | 61.75 | 99.86 |

Values in bold donate traits for the suggested factor name. The full names of the different STIs are listed in Table 1.

The PC2 had a strong positive correlation with PDW-75, BY, and GY measured under the control conditions, SSI, TOL, and RSE calculated based on any plant trait, a strong negative correlation with YSI, a moderate positive correlation with PDW-90 measured under the control conditions, and a moderate negative correlation with SWPI calculated based on PDW-90 and GY (Table 5). Therefore, the different traits and STIs can be classified into two main groups. Group 1 mostly included all traits measured under salinity conditions and the following STIs: STI, YI, REI, SWPI, MRPI, MPI, GMPI, and HMPI, while group 2 mostly included all traits measured under the control conditions and the following STIs: YSI, SSI, TOL, and RSE (Table 5 and Figure 4). Additionally, the vectors of traits and STIs of group 1 formed an acute angle between each other, while those of traits and STIs of group 2 formed an obtuse and straight angle between each other, and with the vectors of all the traits and STIs of group 1 (Figure 4).

According to the PCA biplot, the different genotypes were scattered in the four quarters of the biplot, which indicate a high level of genetic variation among the tested genotypes, and a clear difference existed among salt-sensitive, moderately salt-tolerant, and salt-tolerant genotypes (Figure 4). The salt-tolerant genotypes Kharchia and Sakha 93 were located in the quarter with the highest PC1 and lowest PC2, and closely correlated with all traits measured under salinity conditions and the following STIs: YI, SWPI, and YSI, whereas the salt-sensitive genotype Sakha 61 was located in the opposite quarter (the lowest PC1 and highest PC2) and closely correlated with TOL, SSI, and RSE (Figure 4).

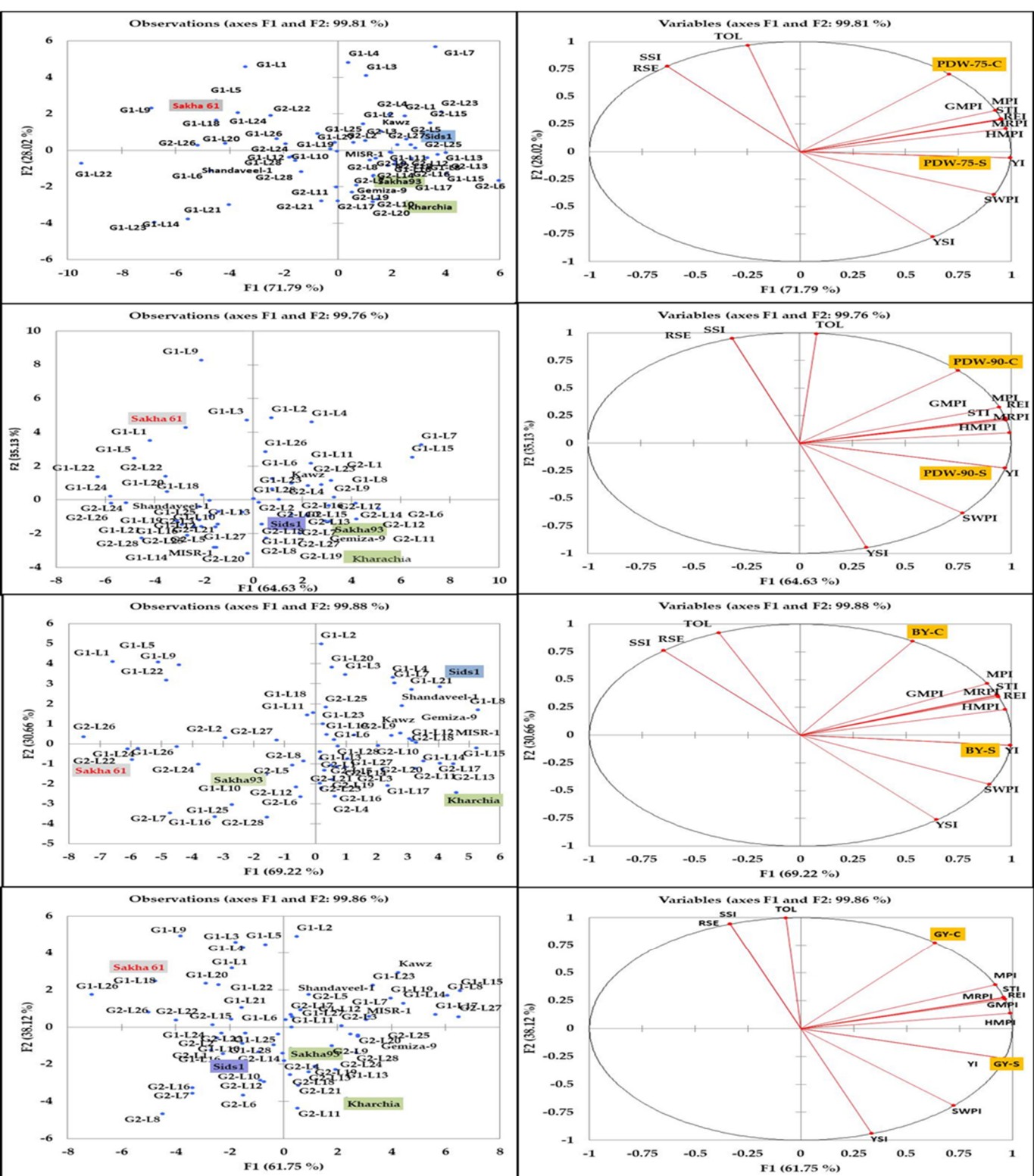

**Figure 4.** Principal component analysis plot showing the first two principal components (PCs) for plant dry weight measured at 75 days after sowing (PDW-75), plant dry weight measured at 90 days after sowing (PDW-90), biological yield (BY), and grain yield (GY) of 64 wheat genotypes grown under the control (C) and salinity stress (S) conditions. The full names of the different STIs are listed in Table 1.

### 3.3.3. Cluster Analysis

Cluster analysis organized the tested genotypes into four major distinct clusters based on the four traits under the control and salinity conditions and the different STIs calculated based on these traits (Figure 5). Cluster 2 included the highest number of genotypes based on PDW-75 (24), BY (22), and GY (30), while cluster 1 included the highest number of genotypes based on PDW-90 (35). The lowest number of genotypes was found in cluster 1 based on PDW-75 (9), cluster 3 based on GY (7), and cluster 4 based on PDW-90 (6) and BY (8) (Table 6). Cluster 1 included the salt-tolerant genotype Kharchia, 1–9 RLIs from group 1, and 9–12 RLIs from group 2, whereas cluster 4 included the salt-sensitive genotype Sakha 61, 5–10 RLIs from group 1, and only 1–2 RLIs from group 2 (Figure 5). Based on PDW-75, GY, and the STIs calculated based on these traits, the salt-tolerant genotype Sakha 93 and moderately salt-tolerant genotype Sids1 with 7–14 RLIs from group 1 and 12–13 RLIs from group 2 were grouped together in cluster 2 whereas, based on BY and their STIs, Sakha 93 with 8 RLIs from group 1 and 10 RLIs from group 2 still stayed in cluster 2, but Sids 1 with 10 RLIs from group 1 and 4 RLIs from group 2 were grouped together in cluster 3. Based on PDW-90, Sakha 93 and Sids 1 were grouped with Kharchia in cluster 1 (Figure 5). The genotypes belonging to cluster 1 attainedhigher values for the four traits measured under salinity conditions and all STIs, except SSI, TOL, and RSE, which attained lower values for these three indices; the opposite was true for the genotypes belonging to cluster 4 (Table 6). The averaged values of the four traits measured under salinity conditions and different STIs of genotypes belonging to cluster 2 were occasionally comparable to those in cluster 1 and cluster 3, and were also occasionally comparable to those in cluster 4 (Table 6). Therefore, the genotypes that formed clusters 1, 2, 3, and 4 can be considered salt-tolerant, moderately salt-tolerant, moderately salt-sensitive, and salt-sensitive genotypes, respectively.

**Table 6.** Comparison profile of the four clusters of 64 wheat genotypes classified by robust hierarchical clustering (cluster figures are means of plant dry weight measured at 75 (PDW-75) and 90 (PDW-90) days after sowing, biological yield (BY), grain yield (GY) under the control (C) and salinity (S) conditions and their different stress tolerance values for the genotypes in each cluster).

| | Cluster 1 | Cluster 2 | Cluster 3 | Cluster 4 | Cluster 1 | Cluster 2 | Cluster 3 | Cluster 4 |
|---|---|---|---|---|---|---|---|---|
| Traits | | PDW-75 | | | | PDW-90 | | |
| No. of genotypes | 9 | 24 | 18 | 13 | 35 | 15 | 8 | 6 |
| C | 5.74 | 5.89 | 6.10 | 6.05 | 7.76 | 8.70 | 7.21 | 9.42 |
| S | 4.55 | 4.36 | 4.16 | 3.51 | 5.63 | 5.69 | 4.33 | 4.78 |
| YSI | 0.79 | 0.74 | 0.68 | 0.58 | 0.73 | 0.66 | 0.60 | 0.51 |
| STI | 0.74 | 0.73 | 0.72 | 0.61 | 0.68 | 0.77 | 0.49 | 0.70 |
| YI | 1.10 | 1.05 | 1.00 | 0.84 | 1.04 | 1.05 | 0.80 | 0.88 |
| REI | 1.06 | 1.05 | 1.04 | 0.88 | 1.02 | 1.15 | 0.72 | 1.04 |
| SWPI | 1.90 | 1.79 | 1.68 | 1.42 | 2.02 | 1.93 | 1.61 | 1.56 |
| MRPI | 2.06 | 2.04 | 2.02 | 1.86 | 2.00 | 2.13 | 1.69 | 2.05 |
| MPI | 5.15 | 5.13 | 5.13 | 4.78 | 6.70 | 7.19 | 5.77 | 7.10 |
| GMPI | 5.11 | 5.07 | 5.03 | 4.60 | 6.61 | 7.03 | 5.58 | 6.70 |
| HMPI | 5.08 | 5.01 | 4.94 | 4.43 | 6.52 | 6.88 | 5.40 | 6.32 |
| SSI | 0.69 | 0.86 | 1.05 | 1.38 | 0.83 | 1.04 | 1.20 | 1.49 |
| TOL | 1.19 | 1.54 | 1.94 | 2.54 | 2.14 | 3.01 | 2.88 | 4.65 |
| RSE | 20.78 | 26.06 | 31.88 | 41.93 | 27.41 | 34.45 | 39.72 | 49.10 |
| Traits | | BY | | | | GY | | |
| No. of genotypes | 18 | 22 | 16 | 8 | 15 | 30 | 7 | 12 |
| C | 17.60 | 18.80 | 20.23 | 18.29 | 5.12 | 5.80 | 5.87 | 5.92 |
| S | 13.79 | 13.74 | 13.13 | 10.00 | 3.97 | 4.03 | 3.76 | 3.30 |
| YSI | 0.78 | 0.73 | 0.65 | 0.55 | 0.78 | 0.70 | 0.64 | 0.56 |
| STI | 0.70 | 0.74 | 0.76 | 0.52 | 0.64 | 0.73 | 0.69 | 0.61 |
| YI | 1.05 | 1.05 | 1.00 | 0.76 | 1.03 | 1.05 | 0.98 | 0.86 |
| REI | 1.00 | 1.06 | 1.09 | 0.75 | 0.94 | 1.08 | 1.02 | 0.90 |
| SWPI | 3.28 | 3.17 | 2.92 | 2.35 | 1.75 | 1.67 | 1.55 | 1.35 |
| MRPI | 1.99 | 2.05 | 2.08 | 1.74 | 1.93 | 2.07 | 2.01 | 1.90 |
| MPI | 15.70 | 16.27 | 16.68 | 14.14 | 4.54 | 4.91 | 4.81 | 4.61 |
| GMPI | 15.57 | 16.07 | 16.28 | 13.49 | 4.51 | 4.83 | 4.69 | 4.41 |
| HMPI | 15.45 | 15.86 | 15.89 | 12.88 | 4.47 | 4.75 | 4.58 | 4.23 |
| SSI | 0.72 | 0.89 | 1.16 | 1.49 | 0.70 | 0.95 | 1.12 | 1.38 |
| TOL | 3.82 | 5.06 | 7.10 | 8.29 | 1.15 | 1.77 | 2.11 | 2.63 |
| RSE | 21.73 | 26.84 | 34.90 | 44.61 | 22.35 | 30.38 | 35.87 | 44.21 |

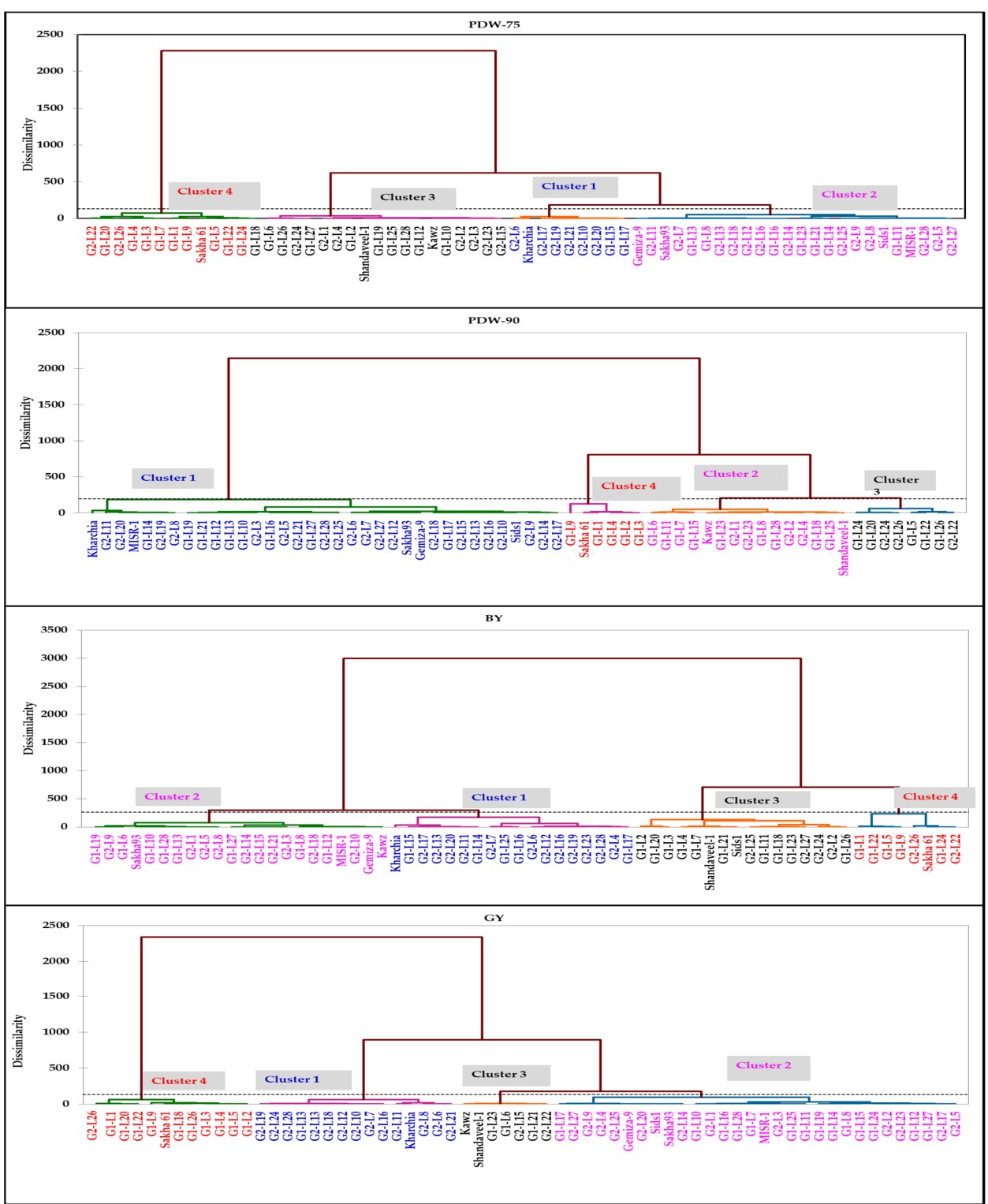

**Figure 5.** Hierarchical clustering derived by Ward's methods of cluster analysis for plant dry weight measured at 75 (PDW-75) and 90 (PDW-90) days after sowing, biological yield (BY), grain yield (GY) and their different stress tolerance indices of 64 wheat genotypes grown under the control and salinity stress conditions. The full names of the different STIs are listed in Table 1. PDW-75, PDW-90, BY, and GY indicate plant dry weight measured at 75 (PDW-75) and 90 (PDW-90) days after sowing, biological yield (BY), and grain yield (GY), respectively. The full names of the different STIs are listed in Table 1.

### 3.4. Ranking of Genotypes for Their Relative Salt Tolerance at Different Growth Stages

The genotypes were ranked for relative salt tolerance at booting (BT, at 75 days after sowing), anthesis (AN, at 90 days after sowing), and maturity (MT) growth stages based on PDW-75 with their STIs, PDW-90 with their STIs, and both BY and GY with their STIs, respectively. The different genotypes were divided into four cluster groups at each growth stage (Table 7). Importantly, the ranking of some genotypes for their relative salt tolerance was changed from one growth stage to another. For example, some genotypes were ranked as salt-tolerant genotypes at the early growth stage, whereas they ranked as moderately salt-tolerant or salt-sensitive genotypes at the AN and MT stages; the opposite was observed with other genotypes (Table 7). Finally, based on the sum ranking of genotypes at different growth stages, Kharchia with 2 RLIs from group 1 and 13 RLIs from group 2, Sakha 93, Sids 1, Gemiza-9, and MISR-1 with 8 RLIs from group 1 and 9 RLIs from group 2, Kawz and Shandaweel-1 with 7 RLIs from group 1 and 4 RLIs from group 2, and Sakha 61 with 11 RLIs from group 1 and 2 RLIs from group 2 were ranked as salt-tolerant, moderately salt-tolerant, moderately salt-sensitive, and salt-sensitive genotypes, respectively (Table 7).

**Table 7.** Rankings of genotypes (Gen.) for their relative salt tolerance at booting (BT), anthesis (AN), and maturity (MT) growth stages based on plant dry weight, biological yield, grain yield, and different stress tolerance indices in cluster analysis (Ward's minimum variance analysis).

| Gen. | BT | AN | MT | Sum | Rank | FTD | Gen. | BT | AN | MT | Sum | Rank | TD |
|---|---|---|---|---|---|---|---|---|---|---|---|---|---|
| Kharchia | 1 | 1 | 1 | 3 | 1 | T | G2-L4 | 3 | 2 | 1 | 6 | 2 | MT |
| G2-L6 | 1 | 1 | 1 | 3 | 1 | T | G2-L25 | 2 | 1 | 3 | 6 | 2 | MT |
| G2-L19 | 1 | 1 | 1 | 3 | 1 | T | G2-L27 | 2 | 1 | 3 | 6 | 2 | MT |
| G2-L10 | 1 | 1 | 1 | 3 | 1 | T | G1-L21 | 2 | 1 | 3 | 6 | 2 | MT |
| G2-L21 | 1 | 1 | 1 | 3 | 1 | T | G2-L15 | 3 | 1 | 2 | 6 | 2 | MT |
| G1-L17 | 1 | 1 | 2 | 4 | 1 | T | G1-L19 | 3 | 1 | 3 | 7 | 3 | MS |
| G2-L11 | 2 | 1 | 1 | 4 | 1 | T | G1-L25 | 3 | 2 | 2 | 7 | 3 | MS |
| G2-L12 | 2 | 1 | 1 | 4 | 1 | T | G2-L23 | 3 | 2 | 2 | 7 | 3 | MS |
| G2-L13 | 2 | 1 | 1 | 4 | 1 | T | G1-L11 | 2 | 2 | 3 | 7 | 3 | MS |
| G2-L16 | 2 | 1 | 1 | 4 | 1 | T | G1-L28 | 3 | 2 | 2 | 7 | 3 | MS |
| G2-L17 | 1 | 1 | 2 | 4 | 1 | T | G2-L1 | 3 | 2 | 2 | 7 | 3 | MS |
| G2-L20 | 1 | 1 | 2 | 4 | 1 | T | Kawz | 3 | 2 | 2 | 7 | 3 | MS |
| G2-L28 | 2 | 1 | 1 | 4 | 1 | T | G1-L23 | 2 | 2 | 3 | 7 | 3 | MS |
| G1-L13 | 2 | 1 | 1 | 4 | 1 | T | G1-L6 | 3 | 2 | 3 | 8 | 3 | MS |
| G2-L8 | 2 | 1 | 1 | 4 | 1 | T | G2-L2 | 3 | 2 | 3 | 8 | 3 | MS |
| G2-L18 | 2 | 1 | 1 | 4 | 1 | T | Shandaweel 1 | 3 | 2 | 3 | 8 | 3 | MS |
| G2-L7 | 2 | 1 | 2 | 5 | 2 | MT | G2-L24 | 3 | 3 | 3 | 9 | 3 | MS |
| G1-L14 | 2 | 1 | 2 | 5 | 2 | MT | G1-L7 | 4 | 2 | 3 | 9 | 3 | MS |
| G1-L15 | 1 | 2 | 2 | 5 | 2 | MT | G1-L18 | 4 | 2 | 4 | 10 | 4 | S |
| G1-L16 | 2 | 1 | 2 | 5 | 2 | MT | G1-L24 | 4 | 3 | 3 | 10 | 4 | S |
| Sakha 93 | 2 | 1 | 2 | 5 | 2 | MT | G1-L26 | 3 | 3 | 4 | 10 | 4 | S |
| Gemiza 9 | 2 | 1 | 2 | 5 | 2 | MT | G2-L22 | 4 | 3 | 3 | 10 | 4 | S |
| MISR 1 | 2 | 1 | 2 | 5 | 2 | MT | G1-L2 | 3 | 4 | 4 | 11 | 4 | S |
| G2-L5 | 2 | 1 | 2 | 5 | 2 | MT | G1-L20 | 4 | 3 | 4 | 11 | 4 | S |
| G2-L9 | 2 | 1 | 2 | 5 | 2 | MT | G1-L5 | 4 | 3 | 4 | 11 | 4 | S |
| G2-L14 | 2 | 1 | 2 | 5 | 2 | MT | G1-L22 | 4 | 3 | 4 | 11 | 4 | S |
| Sids1 | 2 | 1 | 3 | 6 | 2 | MT | G2-L26 | 4 | 3 | 4 | 11 | 4 | S |
| G1-L8 | 2 | 2 | 2 | 6 | 2 | MT | G1-L3 | 4 | 4 | 4 | 12 | 4 | S |
| G1-L10 | 3 | 1 | 2 | 6 | 2 | MT | G1-L4 | 4 | 4 | 4 | 12 | 4 | S |
| G1-L12 | 3 | 1 | 2 | 6 | 2 | MT | Sakha 61 | 4 | 4 | 4 | 12 | 4 | S |
| G1-L27 | 3 | 1 | 2 | 6 | 2 | MT | G1-L1 | 4 | 4 | 4 | 12 | 4 | S |
| G2-L3 | 3 | 1 | 2 | 6 | 2 | MT | G1-L9 | 4 | 4 | 4 | 12 | 4 | S |

T, MT, MS, and S indicate salt-tolerant, moderately salt-tolerant, moderately salt-sensitive, and salt-sensitive genotypes, respectively. FTD indicates the final tolerance degree.

## 4. Discussion

The whole-plant biomass (PDW) is considered one of the most effective screenings and selection criteria for evaluating different genotypes under both stress and non-stress conditions. This is because this plant criterion reflects the response of several physiological and biochemical processes at the whole-plant level to a given environmental condition in a comprehensive manner, reflecting the response of genotypes to given environmental conditions at different phenological growth stages, as well as being an important reserve-storing trait in plants. Therefore, any reduction in PDW caused a significant reduction in the final GY [11,13,47–49]. Moreover, because improving the GY is always the main

target of plant breeders as well as the response of genotypes to abiotic stress varies from one phenological growth stage to another, the measurement of the final GY is also a critical aspect in the evaluation and screening of genotypes under stress conditions [5,13,18,50,51]. In this study, we found a highly significant effect of salinity on PDW measured at 75 (PDW-75) and 90 (PDW-90) days from sowing, BY, and GY (Table 2), with a 30.3, 33.1, 30.0, and 32.1% reduction in these traits under salinity conditions, respectively, when compared with the control treatment (Figure 2). We also observed a highly significant effect of genotypes as well as their interaction with salinity on the four traits (Table 2), with twofold variations found in the four traits among the genotypes under both the control and stress conditions (Figure 2). These results indicate that these traits are effective criteria for evaluating and selecting genotypes under both stress and non-stress conditions. Additionally, the performance of the tested genotypes for the four traits is not consistent across the control and salinity stress conditions. These findings confirm that selection should target genotypes that possess sufficient genetic plasticity to cope with salinity stress, and they should have a relatively high performance across the control and salinity conditions. To achieve this target, different STIs have been suggested as practical tools to follow the performance of the genotypes across contrasting growth conditions. This is because these indices take into account the performance of the genotypes under both stress and non-stress conditions together [14,15,18,24,28]. For example, the YSI is able to recognize the genotypes that have good performance under both stress and non-stress conditions, while the YI is able to identify the genotypes that have good performance under only stress conditions [24,34]. To recognize genotypes that have high yield under stress conditions with high tolerance to stress, the STI is appropriate for achieving this target [30]. Therefore, this index can be successfully used to discriminate between tolerant and sensitive genotypes [52,53]. The genotypes with higher values of SSI are more sensitive to stress and have poor yield stability in both stress and non-stress conditions; therefore, this index is effective to isolate sensitive genotypes [35]. The TOL can recognize the genotypes that attained a low reduction in their biomass and GY under stress conditions compared to non-stress conditions [29]. The genotypes with good performance under non-stress conditions and reasonable performance under stress conditions can be recognized through GMP [36]. Interestingly, the results of this study showed a highly significant variation between tested genotypes for all STIs in both years and combined two years (Table 3), with about two-to-fourfold variations observed in the different STIs among the genotypes (Table 4). These findings indicated that the different STIs could be successfully used as effective criteria to identify the genotypes that produced stably high production of PDW and GY under both stress and non-stress conditions and are highly tolerant to stress. They can also be used to isolate the genotypes that are highly sensitive to stress and produce the lowest PDW and GY under stress conditions and identify the genotypes that produce lower PDW and GY but are tolerant to stress. These findings were confirmed by the significant correlation of the different STIs with the four traits measured under both the control and salinity conditions and with each other (Figure 3). The STI, REI, MRPI, MPI, GMPI, and HMPI exhibited strong and positive correlations with the four traits measured under both the control and salinity conditions, as well as showed strong and positive correlations with each other (Figure 3) which indicated that these indices are able to identify the genotypes that have good performance in both stress and non-stress conditions and, therefore, selecting genotypes with high values for these indices means high PDW and GY under both conditions. The YSI, YI, SWPI, SSI, and RSE exhibited a higher correlation with the four traits measured under salinity conditions than those measured under the control conditions and showed strong and negative correlations with each other (Figure 3), which suggested that these indices are able to identify the genotypes that have good performance under stress conditions and seemed to be effective to discriminate salt-tolerant genotypes from sensitive ones. The opposite situation was found with TOL, which showed a higher correlation with the four traits measured under the control conditions than those measured under salinity conditions (Figure 3), which reflected that

this index is useful to select the genotypes that have good performance under the control conditions only. These results were consistent with the findings of Hajibabaei and Azizi [54] (2011, 54) and Khatibi et al. [55], who reported that GMP, MP, and STI are the indices used most often for selecting the genotypes that have good performance under both stress and non-stress conditions, while SSI is the useful index to evaluate genotypes under severe stress conditions.

*Assessment of the Salt Tolerance of Genotypes Using Multivariate Analysis*

Using MA algorithms, such as the PCA and CA, for the assessment of the salt tolerance of genotypes has several advantages, such as (1) it allows us to evaluate the salt tolerance of genotypes using multiple and various traits; (2) it increases the accuracy of the rankings of genotypes when they are evaluated at different growth stages and across various salinity levels; (3) it allows for the ranking of genotypes easily across salinity levels and different growth stages simultaneously using simple numbers; (4) it divulges complex relationships among the genotypes in a more understandable way; (5) it allows us to identify superior genotypes for both stress and non-stress conditions; (6) it observes the interrelation among traits; and (7) it transforms the number of highly correlated traits into a small number of variables called PCs [19,37,38,56–58]. In this study, the PCA was used to identify the performance of tested genotypes under both the control and salinity conditions using different STIs and the traits measured under both conditions. According to Figure 4, the most variation among all analyzed variables was explained by the first two PCs, with the PC1 and PC2 explained at 61.8–71.8% and 28.0–38.2% of the total variation, respectively. Additionally, the PC1 had a positive and strong correlation with PDW-75 and PDW-90 measured under both conditions, BY and GY measured under salinity conditions, and the following STIs: STI, YI, REI, SWPI, MRPI, MPI, GMPI, and HMPI. Whereas the PC2 had a positive and strong correlation with BY and GY measured under the control conditions, SSI, TOL, and RSE had a negative and strong correlation with YSI (Table 5). These findings indicate that the PC1 is highly correlated with the salt-tolerant indices, growth performance of genotypes under both conditions, as well as yield potential under salinity conditions, while the PC2 is highly correlated with the salt-sensitive indices and yield potential of genotypes under the control conditions. Therefore, the PC1 is able to select the salt-tolerant genotypes as well as the genotypes that perform well under both the salinity and control conditions, while the PC2 is able to isolate the salt-sensitive genotypes, as well as identify the genotypes that only perform well under the control conditions. These results were supported by the scatter of genotypes in the PCA biplot (Figure 4). As shown in this Figure, the salt-tolerant genotypes Kharchia and Sakha 93 are located in the quarter with the highest PC1 and lowest PC2; the opposite was true with the salt-sensitive genotype Sakha 61 (Figure 4). In other words, the salt-tolerant genotypes and promising RILs were located in high yield potential and highly tolerant to salinity, while salt-sensitive genotypes and the RILs that should be isolated from tested germplasm were located in low yield potential under salinity stress conditions and high susceptibility to salinity. These results confirmed the importance of a combination between the PCA and different STIs in classifying the genotypes based on salt tolerance. The use of the PCA combined with STIs to differentiate genotypes for different environmental stresses has also been confirmed by other researchers in different crops and various environmental conditions [23,27,38,59–61].

Cluster analysis also succeeded in classifying genotypes based on their salt tolerance using all STIs and traits measured under the control and salinity conditions simultaneously. This analysis succeeded to group genotypes into four clusters ranging from salt-tolerant to salt-sensitive genotypes; with the most salt-tolerant and salt-sensitive genotypes grouped in cluster 1 and cluster 4, respectively (Figure 5). The salt-tolerant genotypes also attained the highest values for salt-tolerant indices (YI, YSI, REI, SWPI, MRPI, MPI, GMPI, and HMPI), as well as PDW, BY, and GY, particularly under salinity stress conditions; the opposite was true with the salt-sensitive genotypes (Table 6). These findings reveal that



cluster analysis is also effective to differentiate genotypes for salinity stress and the results obtained by the cluster analysis are close to those obtained by the PCA.

Because the salt tolerance of genotypes may change with the growth stage [19], evaluating the salt tolerance of genotypes at various growth stages, as well as across these stages, is important to evaluate genotypes and improve their salt tolerance. The important advantage of cluster analysis (Ward's minimum variance) is their appropriateness to ranking genotypes when they are evaluated at different growth stages. Using this method, the genotypes were ranked at each growth stage by adding the number of Ward's minimum variances at each stage. Thereafter, the genotypes were finally ranked based on the sums of these numbers and the genotypes, with the smallest sums identified as salt-tolerance genotypes, and vice versa. The results in this study show that the salt tolerance of some genotypes was changed from one growth stage to another, but the salt tolerance of the most tolerant and sensitive genotypes was nearly similar at different growth stages (Table 7). These findings indicate that the salt tolerance of salt-tolerant and salt-sensitive genotypes can be detected at one growth stage and the ranking of these genotypes for salt tolerance at this stage was close to that ranking at the other stages. Therefore, such genotypes can be selected at their early growth stages without waiting until the maturity stage, while the salt tolerance of moderately salt-tolerant genotypes and moderately salt-sensitive genotypes should be evaluated at different growth stages.

## 5. Conclusions

Different RILs and genotypes evaluated in this study showed highly significant variation in PDW at different growth stages, BY, and GY, as well as their performance under both the control and salinity stress conditions, which were tested by using different STIs. Therefore, it can be said that the different traits and STIs used in this study may serve as efficient tools to estimate the degree of tolerance to salinity stress in advanced lines and genotypes of wheat. These results have been confirmed when the measured traits and STIs were used together with multivariate analysis (PCA and CA). Based on the results of the PCA, the following STIs: STI, YI, REI, SWPI, MRPI, MPI, GMPI, and HMPI, as well as PDW, BY, and GY under salinity stress conditions, were the best indicators for identifying the salt-tolerant genotypes and the genotypes that perform well under both the control and salinity stress conditions because these traits had strong and positive correlations with the PC1, which explained 61.8–71.8% of the total variation among traits and genotypes. To isolate the most salt-sensitive genotypes, the following STIs: SSI, TOL, RSE, and YSI, were effective. Based on cluster analysis and multiple traits, it is possible to rank wheat genotypes based on their salt tolerance when they are evaluated at different growth stages. Finally, our results confirmed the effectiveness of multivariate analysis that simultaneously applies multiple traits as a powerful explanatory and efficient tool in wheat breeding programs for discriminating the genotypes according to their level of salt tolerance, even at early phenological growth stages.

**Author Contributions:** Conceptualization, S.E.-H. and M.M.; methodology, S.E.-H., M.M., M.U.T., M.A.; N.M., E.T. and Y.R.; software, S.E.-H., M.M. and M.U.T.; validation, S.E.-H.; formal analysis, S.E.-H., M.M., M.U.T., M.A.; N.M., E.T. and Y.R.; investigation, S.E.-H. and M.M.; resources, S.E.-H., M.M. and M.A.; data curation, M.M. and M.U.T.; writing—original draft preparation, S.E.-H.; writing—review and editing, S.E.-H.; visualization, S.E.-H. and M.M; supervision, S.E.-H.; project administration, S.E.-H.; funding acquisition, S.E.-H. All authors have read and agreed to the published version of the manuscript.

**Funding:** This research was funded by the Deputyship for Research & Innovation, Ministry of Education, in Saudi Arabia, grant number IFKSURG-2-1562.

**Institutional Review Board Statement:** Not applicable.

**Informed Consent Statement:** Not applicable.

**Data Availability Statement:** All data are presented within the article.

**Acknowledgments:** The authors extend their appreciation to the Deputyship for Research & Innovation, Ministry of Education, in Saudi Arabia for funding this research work through project No. (IFKSURG-2-1562).

**Conflicts of Interest:** The authors declare no conflict of interest.

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
