# Peer review of "Assessing the Suitability of Multivariate Analysis for Stress Tolerance Indices, Biomass, and Grain Yield for Detecting Salt Tolerance in Advanced Spring Wheat Lines Irrigated with Saline Water under Field Conditions"

_agronomy, doi:10.3390/agronomy12123084_

Round 1
Reviewer 1 Report
The authors gave a nice idea, which is the use of multivariate
analysis as a powerful and effective explanatory tool in wheat breeding programs to distinguish between genotypes according to the level of salt tolerance in the early phenological stages of development.
Can this multivariate analysis be applied to other crops under different levels of salt or drought to prepare new salt and drought tolerant varieties
Author Response
Review #1
The authors gave a nice idea, which is the use of multivariate analysis as a powerful and effective explanatory tool in wheat breeding programs to distinguish between genotypes according to the level of salt tolerance in the early phenological stages of development.
Can this multivariate analysis be applied to other crops under different levels of salt or drought to prepare new salt and drought-tolerant varieties?
Response: The authors greatly appreciate your revisions to our manuscript and support for publishing it. Of course, the multivariate analysis can be applied to other crops under different levels of salt or drought to prepare new salt- and drought-tolerant varieties.

Reviewer 2 Report
Very good work. Please check minor mistakes and read again your manuscript for grammar mistakes

Author Response
Review #2
Very good work. Please check minor mistakes and read again your manuscript for grammar mistakes.
Response: We greatly appreciate your critical observations as well as your constructive and helpful comments. We hope that we could address your questions/comments by the explanations and revisions made in the manuscript. We believe that the manuscript is substantially improved after making the suggested revisions.
- Abstract line 16 “increases” has been changed to “to increase”
- Abstract line 16-17 “increases the accuracy of the genotypes ranking across different growth stages and salinity levels” has been changed into “to facilitate ranking of genotypes across different growth stages and salinity levels”.
- Abstract line 20 “biologically yield (BY)” has been changed into “biological yield (BY)”
- Abstract line 23 “genotypes” has been changed into “genotype”
- Abstract line 31 “is” has been changed into “was”
- Introduction line 49 “toward” has been changed into “towards”
- Introduction line 86 “of plant breeder” has been changed into “of a plant breeder”
- Introduction line 87 “the final GY could be sever also as important criterion for evaluating the salt tolerance of genotypes” has been changed into “ the final GY could also be served as an essential criterion for evaluating the salt tolerance of genotypes”
- Introduction line 95 “identifying” has been changed into “identify”
- Introduction line 95 “is a select of genotypes” has been changed into “to select them”
- Introduction line 97 “in” has been changed into “on”
- Introduction line 113-114 “and therefore the high values of this index indicate more stress tolerant and more potential yield of the genotype” has been changed into “and therefore the high values of this index indicate a high yield potential with a high tolerance to stress for a given genotype”
- Introduction line 117 “bio-mass” has been changed into “biomass”
- Introduction line 134 “a” has been changed into “the”
- Introduction line 144 “The second group included genotypes had a lower GY” has been changed into “The second group included genotypes which had a lower GY”
- Introduction line 147 “the third group included genotypes had stable” has been changed into “the third group included genotypes that had stable”
- M&M line 165-166: the sentence has been rewritten as following “the parents and commercial cultivars were previously evaluated under salinity conditions and Sakha 61 and Shandawel-1 have been identified as salt-sensitive, Misr-1 has been identified as moderately salt-sensitive, Sids 1 and Gemiza 9 have been identified as moderately salt-tolerant, and Sakha 93 and Kharchia 65 have been identified as salt-tolerant cultivars.
- M&M line 177 “is very hot and dry days” has been changed into “is with very hot and dry days”
- M&M line 185 “The experiment was conducted” has been changed into “ Field experiments were conducted”
- M&M line 187-188 “After the studied farm was” has been changed into “After the experimental site was”
- M&M line 191 “consists” has been changed into “consisted”
- M&M line 221 “till” has been changed into “to”
- M&M line 223 “control” has been changed into “the control”
- M&M line 227 “The artificial saline” has been changed into “The artificial saline water”
- M&M line 228 “consists” has been changed into “consisted”
- M&M line 231 “for control” has been changed into “for the control”
- Results line 315 “were” has been changed into “are”
- Results line 320 “ranged” has been changed into “ranging”
- Results line 355 “two-four times” has been changed into “2 to 4 times”
- Results line 482-484 “had” has been deleted
- Results line 520 “belong to cluster 4” has been changed into “belonging to cluster 4”
- Results line 523 “cluster” has been changed into “clusters”.

Reviewer 3 Report
Dear appreciated Authors,
Author's paper should be accepted for publication, after minor revisions, since the paper represents contribution to the science, as well as for wheat production on less productive, saline soil.
However, there are a few problems should be considered (see below) and some revisions have to be made and before it can reach a publishable value.
Line 42-43: Today, the world is facing the worst food crisis, while drastic and unexpected changes in the climate are predicted to have a wide range of detrimental effects on global food security. (There is a not appropriate place to mention COVID and Ukraina)
Line 87: serve (instead of sever)
Line 133-134: Multivariate analysis (MA), including principal component analysis (PCA) and cluster analysis (CA) represents one of the most appropriate analysis for the assessment of crop stress performance. This analysis provide identication the most important plant traits governing stress tolerance, as well as discrimination the most stress-tolerant genotypes in process of the screening genotypes for stress tolerance within different salinity levels, and different growth stages.
Line 157: The results of this study could provide the identification of suitable wheat genotypes which can be successfully used in wheat production on less productive, saline framers fields.
Line 264-286: The following steps of data nalysis were performed on PDW-75, PDW-90, BY, GY, and different STIs in order to detect the salt tolerance of evaluated wheat genotypes. To assess the impact of salinity, genotypes, and their interactions on the different plant traits, the values of each year were subjected to analysis of variance (ANOVA) appropriate for split-plot design according to a completely randomized design. Prior to analysis, the normality of each plant trait was tested using the Shapiro-Wilk test. A box plot was used to present the descriptive statistics of the four traits. Pearson correlation matrix was used to For the estimation the correlation level between plant traits and different STIs and between each other the Pearson correlation was used.
To identify the traits that contributed most of the variation in tested wheat genotypes, to detect the interrelationships among multiple traits of each other, as to identify which genotypes were more tolerant of sensitive to salinity stress, PCA was applied to the genotype by a trait matrix of means and and a biplot was drawn using XLSTAT package. To group genotypes according to the level of salt tolerance, a hierarchical cluster analysis was performed at each growth stage based on different STIs and plant trait measured under control and salinity conditions. The analysis was performed according to the Ward’s method, where the distances between the two clusters were expressed as squared Euclidean distances. A dendrogram of clusters was performed to identify the cluster groups, whereas the K-means analysis was used to identify the number of clusters. Ranking of genotypes for salt tolerance at each growth stage and across different growth stages were performed according to the methods of El-Hendawy et al. [19].
Line 290: measured traits
Line 593-597: This is because this plant criterion reflects the response of several physiological and biochemical processes at the whole plant level to a given environmental condition in a comprehensive manner, reflects the response of genotypes to given environmental conditions at different phenological growth stages. Since it is an important reserve-storing trait in plants, any reduction in PDW can cause a significant reduction in the final GY [11,13,47–49]
Best regards,

Author Response
Review #3
Author's paper should be accepted for publication, after minor revisions, since the paper represents contribution to the science, as well as for wheat production on less productive, saline soil. However, there are a few problems should be considered (see below) and some revisions have to be made and before it can reach a publishable value.
Response: We greatly appreciate your critical observations as well as your constructive and helpful comments. We hope that we could address your questions/comments by the explanations and revisions made in the manuscript. We believe that the manuscript is substantially improved after making the suggested revisions.
- Line 42-43: Today, the world is facing the worst food crisis, while drastic and unexpected changes in the climate are predicted to have a wide range of detrimental effects on global food security. (There is a not appropriate place to mention COVID and Ukraina)
Response: Thank you very much for your comment. The sentence has been changed accordingly.
- Line 87: serve (instead of sever)
Response: The author would like to thank the reviewer for the suggestion. “serve” has been changed into “served”.
- Line 133-134: Multivariate analysis (MA), including principal component analysis (PCA) and cluster analysis (CA) represents one of the most appropriate analysis for the assessment of crop stress performance. This analysis provide identication the most important plant traits governing stress tolerance, as well as discrimination the most stress-tolerant genotypes in process of the screening genotypes for stress tolerance within different salinity levels, and different growth stages.
Response: The author would like to thank the reviewer for the suggestion. The sentence has been rewritten as following: Multivariate analysis (MA) including principal component analysis (PCA) and cluster analysis (CA) is the most appropriate and successful analysis that makes the assessment of crop stress performance more practical and reliable, identifies the most important plant traits governing stress tolerance, and discriminates the most stress-tolerant genotypes when the screening of genotypes for stress tolerance was made on multiple traits, different salinity levels, and different growth stages.
- Line 157: The results of this study could provide the identification of suitable wheat genotypes which can be successfully used in wheat production on less productive, saline framers fields.
Response: The author would like to thank the reviewer for the suggestion. The sentence has been changed accordingly.
- Line 264-286: The following steps of data analysis were performed on PDW-75, PDW-90, BY, GY, and different STIs in order to detect the salt tolerance of evaluated wheat genotypes. To assess the impact of salinity, genotypes, and their interactions on the different plant traits, the values of each year were subjected to analysis of variance (ANOVA) appropriate for split-plot design according to a completely randomized design. Prior to analysis, the normality of each plant trait was tested using the Shapiro-Wilk test. A box plot was used to present the descriptive statistics of the four traits. Pearson correlation matrix was used to for the estimation the correlation level between plant traits and different STIs and between each other the Pearson correlation was used. To identify the traits that contributed most of the variation in tested wheat genotypes, to detect the interrelationships among multiple traits of each other, as to identify which genotypes were more tolerant of sensitive to salinity stress, PCA was applied to the genotype by a trait matrix of means and and a biplot was drawn using XLSTAT package. To group genotypes according to the level of salt tolerance, a hierarchical cluster analysis was performed at each growth stage based on different STIs and plant trait measured under control and salinity conditions. The analysis was performed according to the Ward’s method, where the distances between the two clusters were expressed as squared Euclidean distances. A dendrogram of clusters was performed to identify the cluster groups, whereas the K-means analysis was used to identify the number of clusters. Ranking of genotypes for salt tolerance at each growth stage and across different growth stages was performed according to the methods of El-Hendawy et al. [19].
Response: The author would like to thank the reviewer for the suggestion. The paragraph of Analysis of Data has been changed accordingly.
- Line 593-597: This is because this plant criterion reflects the response of several physiological and biochemical processes at the whole plant level to a given environmental condition in a comprehensive manner, reflects the response of genotypes to given environmental conditions at different phenological growth stages. Since it is an important reserve-storing trait in plants, any reduction in PDW can cause a significant reduction in the final GY [11,13,47–49]
Response: The author would like to thank the reviewer for the suggestion. The paragraph of Analysis of Data has been changed accordingly.
